# MineAnyBuild: Benchmarking Spatial Planning for Open-world AI Agents

**Ziming Wei[1]\*, Bingqian Lin[2]\*, Zijian Jiao[1]\*, Yunshuang Nie[1], Liang Ma[3]**
**Yuecheng Liu[4], Yuzheng Zhuang[4], Xiaodan Liang[1]†**

[1]Shenzhen Campus of Sun Yat-sen University   [2]Shanghai Jiao Tong University
[3]Mohamed bin Zayed University of Artificial Intelligence   [4]Huawei Noah's Ark Lab

Project Website: `https://mineanybuild.github.io/`

## Abstract

Spatial Planning is a crucial part in the field of spatial intelligence, which requires the understanding and planning about object arrangements in space perspective. AI agents with the spatial planning ability can better adapt to various real-world applications, including robotic manipulation, automatic assembly, urban planning *etc*. Recent works have attempted to construct benchmarks for evaluating the spatial intelligence of Multimodal Large Language Models (MLLMs). Nevertheless, these benchmarks primarily focus on spatial reasoning based on typical Visual Question-Answering (VQA) forms, which suffers from the gap between abstract spatial understanding and concrete task execution. In this work, we take a step further to build a comprehensive benchmark called **MineAnyBuild**, aiming to evaluate the spatial planning ability of open-world AI agents in the *Minecraft* game. Specifically, MineAnyBuild requires an agent to generate *executable architecture building plans* based on the given multi-modal human instructions. It involves 4,000 curated tasks and provides a paradigm for infinitely expandable data collection by utilizing rich player-generated content. MineAnyBuild evaluates spatial planning through four core supporting dimensions: spatial understanding, spatial reasoning, creativity, and spatial commonsense. Based on MineAnyBuild, we perform a comprehensive evaluation for existing MLLM-based agents, revealing the severe limitations but enormous potential in their spatial planning abilities. We believe our MineAnyBuild will open new avenues for the evaluation of spatial intelligence and help promote further development for open-world AI agents capable of spatial planning.

## 1   Introduction

Spatial intelligence, an emerging research field gradually attracting the attention of AI researchers, requires AI agents to understand, reason and memorize the visual-spatial relationships between objects and spaces [1, 2, 3, 4]. Spatial planning is a pivotal capability regarding spatial intelligence, which requires agents to not only perform spatial perception and cognition, but also generate executable planning in 3D space. Spatial planning is widely needed in various human-centric real-world applications, including automatic assembly, architectural design, environmental urban planning, *etc*.

AI Agents integrated with Multi-modal Large Language Models (MLLMs) have demonstrated astonishing capabilities in various tasks in text (1D) and image (2D) domains [5, 6, 7]. To investigate how existing MLLM-based agents can handle space dimension tasks, several benchmarks designed for evaluating the spatial intelligence have been proposed recently [8, 9, 10, 11]. These benchmarks reveal that although AI agents perform well in tasks of text and image domains, they still present

---

\*Equal Contribution. †Corresponding Author.

39th Conference on Neural Information Processing Systems (NeurIPS 2025) Track on Datasets and Benchmarks.

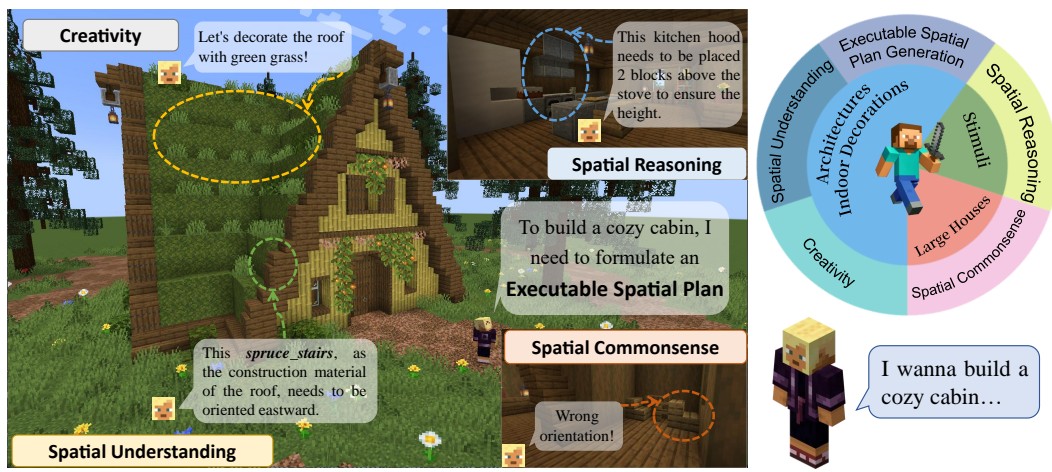

Figure 1: Overview of **MineAnyBuild**. Our MineAnyBuild is a novel benchmark built on the Minecraft game, which aims to evaluate the spatial planning capabilities of open-world AI agents. In MineAnyBuild, the agent needs to generate *executable spatial plans* to construct a building or indoor decoration following given multi-modal human instructions. We introduce four core dimensions, including spatial understanding, creativity, spatial reasoning, and spatial commonsense to fulfill a comprehensive assessment for spatial planning.

relatively poor performance in spatial dimension tasks. However, current benchmarks have critical constraints in evaluating spatial intelligence. They mainly focus on metric-level spatial understanding tasks and predominantly employ Visual Question-Answering (VQA) pairs, requiring AI agents to answer geometric attributes (e.g., distance, positional coordinates, or spatial relations of objects in 3D space), while neglecting the gap between abstract spatial understanding and concrete task execution.

In this work, we propose **MineAnyBuild**, which is an innovative benchmark designed to evaluate an important yet unexplored aspect of spatial intelligence, i.e., spatial planning, for open-world AI agents. In our MineAnyBuild, the agents need to generate *executable spatial plans* following human instructions for constructing a building or indoor decoration, which requires both spatial reasoning and task execution. We build our benchmark on the popular *Minecraft* game, where a player journeys through a 3D world with diverse biomes to explore, tools to craft, and architectures to build. Compared to benchmarks focusing on skills learning or tech-tree tasks [12, 13], architecture building has always been a vital attraction to millions of players to present the openness and freedom of Minecraft. Unlike most other games, Minecraft defines go-as-you-please goals, making it well suited for developing open-ended tasks for AI agent research.

Our MineAnyBuild benchmark consists of 4,000 curated tasks where four core evaluation dimensions are introduced. As shown in Figure 1, given a multimodal human instruction, agents are requested to perform **spatial understanding** to abstract a pivotal basic structure according to specific or brief demands, where agents emulate architects in our real world, and plan the composition of each basic units. Agents also need to reason and think about whether the units of the architecture from different perspectives conforms to spatial rules by **spatial reasoning**. For the overall appearance of the architecture, agents exert their **creativity** and imagination to make it more aesthetically unique, or to simulate some well-designed or delicate structural designs in the real world through the combination of fixed-shaped blocks, e.g., using a variety of stairs and slabs to design unique Chinese-style or castle-style roofs. For some architectures like modern houses, agents implement **spatial commonsense** to judge the rationality of each designs inside the buildings.

We design and construct different tasks based on multiple aspects that a human player would consider, to evaluate several capabilities of AI agents. Agents are supposed to response with concrete layout sequences of expected architectures to present their spatial planning. For some tasks that are not easy to evaluate directly like spatial reasoning, we customize tasks inspired by classical mental rotation experiments [14, 15] to test agents. For creativity, we score and vote on the overall aesthetics and structural strategy by human evaluation or critique-based MLLMs. We also propose an infinitely expandable paradigm to utilize Minecraft data on the Internet, where millions of active players provide their creation and shares, to build our tasks. Through our data curation pipeline, we can collect

endless tasks evaluating spatial intelligence for open-world agents, making further contributions to promoting AI agents research.

We test the tasks on several state-of-the-art MLLM-based AI agents and observe that even the most powerful MLLMs like GPT-4o and Claude-3.7-Sonnet demonstrate significant limitations in most tasks, where GPT-4o obtains an overall score of 41.02, far lower than the maximum score of 100. Open-source models generally have poor capabilities to generate executable spatial plan, reflecting a serious deficiency in their understanding of spatial data. These results reveal the foresight of our MineAnyBuild for AI evaluation.

To summarize, the main contributions of this work are as follows:

- We propose MineAnyBuild, which benchmarks the spatial planning evaluation for open-world AI agents in the Minecraft game. MineAnyBuild covers diverse evaluation dimensions, including spatial reasoning, creativity, spatial commonsense, *etc*. Through requiring the agent to generate executable architecture building plans, our MineAnyBuild significantly mitigate the gap between abstract spatial understanding and concrete task execution.

- We test various existing MLLM-based AI agents for spatial planning in multiple perspectives and difficulties, which exposes the insufficiency of the existing AI agents' capabilities in spatial planning. We provide the visualization results on executable planning outputs and failure cases, revealing that current AI agents are still facing tough issues such as spatial misunderstanding and implementation gap to be handled.

- We propose an infinitely expandable data curation pipeline to scale our benchmark and datasets, where we can collect endless player-generated content on the Internet and automatically convert it into processable data. Our pipeline well utilize the abundant creations made by millions of players to benefit the training and evaluation of open-world AI agents.

## 2 Benchmark and Task Suite

In this section, we describe our MineAnyBuild benchmark in detail. Specifically, we first present the overview of our benchmark in Section 2.1. Then, we define various spatial planning tasks in MineAnyBuild in Section 2.2. Finally, we introduce our data curation pipeline in Section 2.3.

### 2.1 Benchmark Overview

MineAnyBuild is designed to evaluate an AI agent's capabilities in spatial planning to conduct infinite architecture creations in Minecraft game. Spatial planning is a critical capability, aiming to examine agents' understanding and disassembly of combinations in 3D space, and to construct each sub-units and judge the rationality of them by reasoning or commonsense. Our benchmark examines various MLLM-based agents to conduct planning on architectures, which requires them to generate executable architectural construction plans according to different forms of instructions or visual inputs. Evaluating the creativity of agents becomes essential in our particular architectural construction tasks, as it reflects human-centric assessments of aesthetic value across spatial planning and conception designing domains. For some evaluating dimensions that are not easily presented directly in spatial planning, such as spatial reasoning and spatial commonsense, we design series of visual question-answering pairs to indirectly reflect the manifestation of these two capabilities of agents. The next sections detail the specific tasks and the process of our data curation.

### 2.2 Tasks

Our MineAnyBuild involves approximately 4,000 spatial planning tasks with 500+ buildings/indoor decoration assets. These tasks, including Executable Spatial Plan Generation, Spatial Understanding, Creativity, Spatial Reasoning, and Spatial Commonsense, correspond to diverse evaluation dimensions, thereby conducting a comprehensive assessment of AI agents' spatial planning capabilities. In Executable Spatial Plan Generation, Spatial Understanding, and Creativity tasks, the agent needs to generate executable spatial plans for building an architecture according to the given instruction. While in Spatial Reasoning and Spatial Commonsense tasks, we introduce ∼2,000 VQA pairs, where we ask the agent to answer the given questions accompanied by the related images. In the following, we define each task in detail, and present the corresponding task examples in Figure 2.

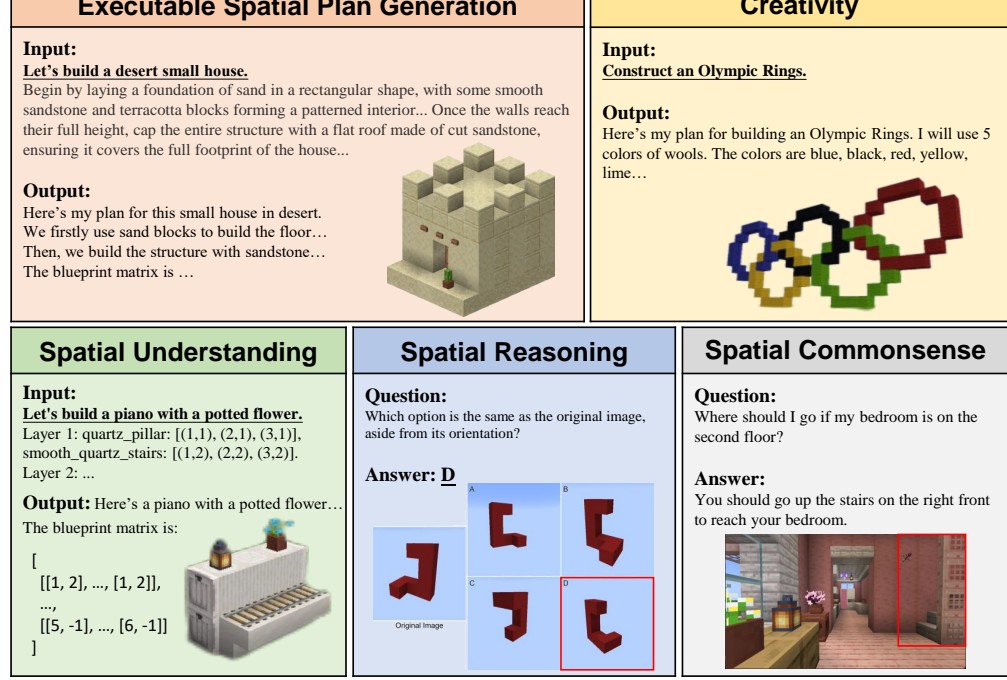

Figure 2: Task examples of **MineAnyBuild**. We present five task examples with specific inputs (questions) and outputs (answers). Some of them are simplified to illustrate the core presentations.

**Executable Spatial Plan Generation.** To construct an architecture, an agent first needs to design the layout of the architecture and accordingly generate the executable spatial plans based on its spatial perception, spatial understanding, and abundant knowledge. Based on this motivation, we propose the Executable Spatial Plan Generation task, which evaluates agents' abilities to perform Spatial Planning. The task input is an abstract architecture building instruction accompanied by precise explanations. Under the given task input, the agents are required to think on the decomposition of architecture substructures and corresponding connections to generate executable spatial plans for architecture building, just like completing a jigsaw puzzle.

For example, in this task instruction "*Build an apple...The apple also needs to have a stem, which we can use black_terracotta to make it.*" for architecture building, we lead the agent to think on the *stem* substructure in *apple* architecture and how to connect it with other substructures. If the agent could understand and plan in spatial perspective, the result should be better than planning in an abstract perspective. For the instruction regarding the indoor decoration, the agents are challenged to make more delicate and exquisite design and planning. More details of the Executable Spatial Plan Generation task are provided in the Supplementary Material.

**Spatial Understanding.** Inspired by the popular instruction following tasks [16, 17, 18] which are widely developed for evaluating MLLM-based agents, we introduce a Spatial Understanding task, where the agent needs to build the architecture according to the step-by-step instruction containing the positions of each building block through abstract spatial understanding. Specifically, we label the parts of our data with ground-truth annotations and generate the instructions with a mapping table of relative coordinate corresponding to the pivot position, for instance, *Layer 2: "red_wool": [(0,0),(1,0)]...*, where the block types and relative positions are provided. The agents are required to translate it into a blueprint matrix, which reflects the cognitive transition and integration of relative and holistic spatial understanding, simulating human-like cognitive mapping mechanisms that dynamically balance egocentric (body-centered) and allocentric (world-centered) perspectives.

**Creativity.** Architecture constructing and indoor decoration designing are attracting evaluation tasks than the previous tasks [12, 19, 20, 21, 22], like textual reasoning and coding. In our MineAnyBuild, we introduce a novel Creativity task for evaluating architecture building, where agents receive an instruction and are required to brainstorm block combinations for different parts of the architecture and outline a rough structure layout, to find ways to maximize creativity and the dynamic range of possible builds.

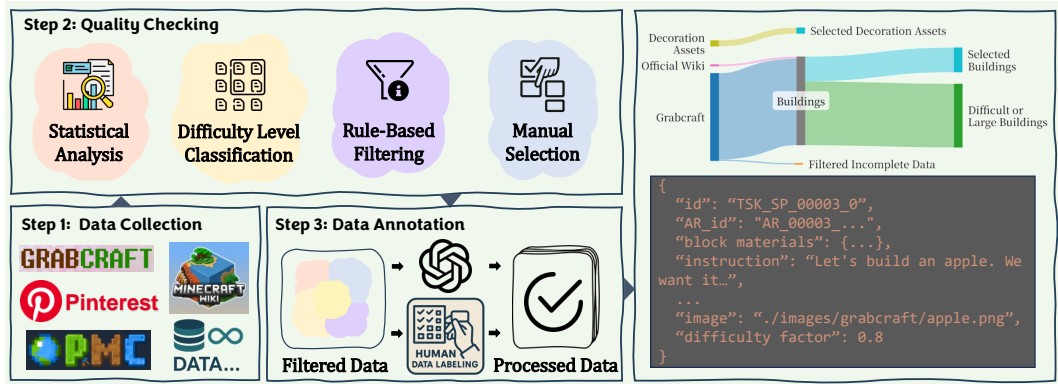

Figure 3: Data curation pipeline of **MineAnyBuild**. We conduct three core steps to curate our datasets: data collection, quality checking, and data annotation. On the right side, a Sankey diagram showing our data processing flow is presented, along with an example of simplified format of processed data.

It is worth elaborating that although this evaluation dimension is different from other traditional ones, creativity is an aesthetic and humanistic criterion that is more in line with human intuition and consensus, akin to the aesthetic assessment of image generation tasks. Creativity is a crucial component of scientific thinking that reflects the process of new things replacing old ones, which is precisely the spirit that scientific researchers have always pursued. Therefore, creativity not only reflects the cognitive depth of future AI systems, but also emerges as a novel and important criterion for agents towards Artificial General Intelligence (AGI). We test creativity through state-of-the-art MLLM-based critic models and human evaluations. Despite the challenges in standardizing evaluation criteria and achieving inter-rater reliability, the majority of recognized favorable comments or votes indicate that agents exhibit measurable creativity.

**Spatial Reasoning.** Spatial reasoning is the ability to imagine, visualize and differentiate objects in 3D space [5, 23, 24]. Inspired by the classic experiments in psychology named *mental rotation* [25, 15, 26], we construct 48 geometric objects made of blocks, denoted as *stimuli*, and generated 1,900 tasks for evaluation of spatial reasoning. As shown in Figure 2, for example, the agent needs to analyze the geometric structure of the given stimuli and determine whether others are the same or not as the given one. We conduct these tasks with Visual Question-Answer pairs, which is easier and more accurate for evaluation.

**Spatial Commonsense.** Spatial commonsense is the critical intuitive comprehension in our daily life that humans possess about the spatial attributes of objects in the physical world, including location, orientation, distance, shape, *etc.* [27, 28, 29]. Spatial commonsense is generally reflected in several aspects: 1) navigation and sense of direction: human can roughly orient the bedroom without a map. 2) rationality of object placement: a refrigerator cannot be placed in a bathroom. We evaluate these tasks on agents and we place the complete commonsense tasks tested in the Supplementary Material.

## 2.3 Data Curation

MineAnyBuild is a comprehensive benchmark with diverse architectures and indoor decorations, aligned with various instructions and visual reference images. We build our benchmark based on the following steps: 1) data collection, 2) quality checking, and 3) data annotation. Figure 3 presents our data curation pipeline for constructing MineAnyBuild.

**Data Collection.** Benefiting from the abundant and creative player-generated content on the Internet, we first collect ∼7000 architectures from several websites, e.g., GrabCraft [30] and Minecraft Official Wiki [31, 32], and collect ∼500 indoor decoration assets from sharing platforms by Minecraft creators. For some player-uploaded data containing potential issues, we filter out these problematic data with some quality standards. For spatial reasoning tasks, we collect 48 stimuli referencing the classic mental rotation experiments [14] and generate three groups of chiral stimuli symmetrical about the X/Y/Z coordinate axes. We design three major types of questions to construct VQA data for the spatial reasoning task based on these generated stimuli. These questions involve having agents select the only one among the four options that differs from or is the same as the stimulus in reference image, or to determine whether the stimuli in the two images are consistent. We utilize the data of

large-scale buildings incorporating interior decorations to generate VQA pairs that complies with spatial commonsense, and further request agents to plan how to construct these decoration assets.

**Quality Checking and Data Annotation.** We implement some codes to filter the problematic data, and then we conduct a human review process to maintain high quality for data annotation. We annotate the instructions of tasks by human or state-of-the-art MLLMs. Specifically, we first carefully design some instructions that guide the agents to think about the decomposition and construction of the architectures, thereby more closely aligning with the motivation of spatial planning. For spatial commonsense tasks, we manually design the VQA pairs that well-fit with questions in the real world.

**Infinitely Expandable Paradigm.** As shown in Figure 3, we provide an infinitely expandable paradigm for data curation, facilitating the subsequent development of training and evaluation resources to advance AI agents research for spatial planning. Through our infinitely expandable paradigm, we can collect the majority of the existing player-generated content on the Internet and import it into the Minecraft game. Specifically, we manually mark the starting block (the minimum values on the X/Y/Z coordinates) and the ending block (the maximum values on the X/Y/Z coordinates) of the 3D coordinates as the three-dimensional coordinate box of the entire building, and obtain all the block information corresponding to each position through *mineflayer* simulator [33]. After filtering the *"air"* blocks, corresponding *three_d_info*, *blueprint* and *block_materials* can be acquired, with which we can generate this building by calling high-level commands in a blank Minecraft environment and obtain the corresponding visual images through manual screenshot for MLLM or manual annotation. The data that finally generate follows the requirements of our datasheet in terms of format, ultimately ensuring that all data has its unified format.

# 3 Experiments

In this section, we describe the agents evaluated on MineAnyBuild and corresponding evaluation metrics, followed by the results and analysis of performance of agents on MineAnyBuild.

## 3.1 Agents

We mainly conduct our evaluation on MLLM-based agents that suitable to address the spatial planning task in our benchmark. To adapt MLLM-based agents to our spatial planning task, we ask the agents to directly output the executable blueprint matrices. Then, the matrices are subsequently utilized by *mineflayer* simulator [33] to automatically generate corresponding architectures in Minecraft environment. We evaluate 13 MLLMs for our MineAnyBuild. For proprietary models, we evaluate Claude-3.5-Sonnet, Claude-3.7-Sonnet [34], Gemini-1.5-Flash, Gemini-1.5-Pro, Gemini-2.0-Flash [7], GPT-4o, GPT-4o-mini [35]. For open-source models, we evaluate InternVL2.5-[2B/4B/8B] [36], Qwen2.5VL-[3B/7B] [37], LLava-Onevision-7B [38].

All evaluations are conducted in a zero-shot manner for a fair comparison. We also provide RL-based agents for future research and adaptation. We place the detailed information of them and compute resources for agents in the Supplementary Material.

## 3.2 Evaluation Metrics

We introduce diverse metrics for evaluating different tasks in our benchmark. For the Executable Spatial Plan Generation, Creativity, and Spatial Commonsense tasks, as their results do not have a definitely correct or perfect answer, we use the state-of-the-art MLLM (GPT-4.1 [39]) as the critic model to score the planning. Specifically, we query GPT-4.1 to score separately based on different evaluation sub-dimensions and calculate a weighted "Evaluation Score". For different tasks, we obtain a comprehensive score based on the score, denoted as **"Score" (out of 10)** shown in Table 1, through corresponding weighting to indicate the performance of agents in each task. For some cases where the plans generated by agents are not executable, we directly set the scores of these cases to 0, showing that agents have failed in these cases. For the Spatial Reasoning task, we directly calculate the **Accuracy(%)** of the agent's responses as our results. More details about the evaluation metrics and the weighted formulas of scores are given in the Supplementary Material.

Table 1: Evaluation results of AI agents on **MineAnyBuild**. Gray indicates the best performance of each evaluation dimension among all agents and Light Gray indicates the second best results. We also highlight the top three agents based on their overall performance with Dark Orange , Orange , Light Orange , respectively.

| Models | Executable Spatial Plan Generation | Spatial Understanding | Spatial Reasoning | Creativity | Spatial Commonsense | Overall |
|---|---|---|---|---|---|---|
| | Score ↑ | Score ↑ | Accuracy ↑ | Score ↑ | Score ↑ | |
| *Proprietary* | | | | | | |
| Claude-3.5-Sonnet | 3.21 | 4.63 | 19.8 | 3.24 | 6.90 | 39.92 |
| Claude-3.7-Sonnet | 3.48 | 5.07 | 17.6 | 3.10 | 6.94 | 40.70 |
| Gemini-1.5-Flash | 2.87 | 2.49 | 25.8 | 2.71 | 7.12 | 35.54 |
| Gemini-1.5-Pro | 3.53 | 4.80 | 16.9 | 2.73 | 7.52 | 40.54 |
| Gemini-2.0-Flash | 2.63 | 4.19 | 16.0 | 2.44 | 6.82 | 35.36 |
| GPT-4o | 3.27 | 4.75 | 24.4 | 2.73 | 7.32 | 41.02 |
| GPT-4o-mini | 2.08 | 2.52 | 26.7 | 2.38 | 7.14 | 33.58 |
| *Open-source* | | | | | | |
| InternVL2.5-2B | 0.24 | 0.34 | 19.8 | 0.28 | 4.94 | 15.56 |
| InternVL2.5-4B | 0.32 | 0.42 | 20.0 | 0.63 | 5.66 | 18.06 |
| InternVL2.5-8B | 0.68 | 0.62 | 20.4 | 0.66 | 5.62 | 19.24 |
| Qwen2.5VL-3B | 0.63 | 0.61 | 17.0 | 0.54 | 5.46 | 17.88 |
| Qwen2.5VL-7B | 1.29 | 1.12 | 16.0 | 1.34 | 6.30 | 23.30 |
| LLava-Onevision-7B | 0.73 | 0.92 | 19.6 | 0.98 | 5.54 | 20.26 |

## 3.3 Results Analyses

We include evaluation results of tested agents and Output Success Rate (OSR) in Table 1 and Figure 4, respectively. We also provide some output results and failure cases in Figure 5 for specific analyses.

**Task Performance Results.** We evaluate 13 MLLM-based agents on our MineAnyBuild benchmark, including 7 proprietary models and 6 open-source models. From Table 1, we can see that for most proprietary models, the performances are much better than those of open-source models. However, these proprietary models still perform relatively poorly in terms of the average absolute scores, e.g., even the GPT-4o with the highest overall score of 41.02 achieves less than half of the full score of 100. We analyze the task-specific findings as follows:

(1) **Executable Spatial Plan Generation**: For some low-parameter MLLM-based agents (e.g. InternVL2.5-2B/4B), they tend to understand the basic elements in the given instruction or image, and offer a simple or detailed plan for constructing. However, they often encounter difficulties when generating the executable spatial plan and cannot convert their planning into an executable 3D matrix, thus leading to scores under 0.4 as shown in Table 1. For some large-parameter models, they can usually understand the block materials partly compared to low-parameter ones, and can actively select some diverse blocks to build structures. Nevertheless, they often fail to understand the correlations between various combinations of block materials, resulting in a faulty completion of the final building and thus poor scores (from 0.63 to 1.29 in Table 1) by critic model. Proprietary models often achieve a relatively good balance in this aspect, but their planning is generally limited to a boxy or conservative design, and therefore their creations are not highly appreciated by the critic model with the average score of 3.01 ultimately.

(2) **Creativity**: Most proprietary MLLM-based agents can leverage their imagination to construct relatively novel designs, but their 3D architectural capabilities are weak, leading to the difficulty in outputting the executable plans corresponding to their planning and design. Conversely, open-source MLLMs receive lower scores frequently due to their invalid output results rather than the creative plans they generate, yielding a maximum score of 1.34 as quantified in Table 1.

(3) **Spatial Understanding**: In Table 1, the majority of proprietary models achieve solid results, while Gemini-1.5-Flash frequently generates matrices with more than three dimensions resulting execution errors, which suggests a limited grasp of structural understanding. For open-source models, they struggle with the building structures and mainly respond with repeated or increasing matrix results shown in Figure 5, which points to unclear interpretations of task goals.

(4) **Spatial Reasoning**: Spatial reasoning tasks, i.e., mental rotation experiments, require agents to simulate how humans' brain recognizes and moves the stimuli by rotating 3D objects in the mental representation based on the reference stimulus. The distractors are generally mirror-reversed geometries of the stimuli with extra rotations to increase task difficulty. From Table 1, we can observe that most MLLM-based agents perform poorly on this task, where even the top-performing model,

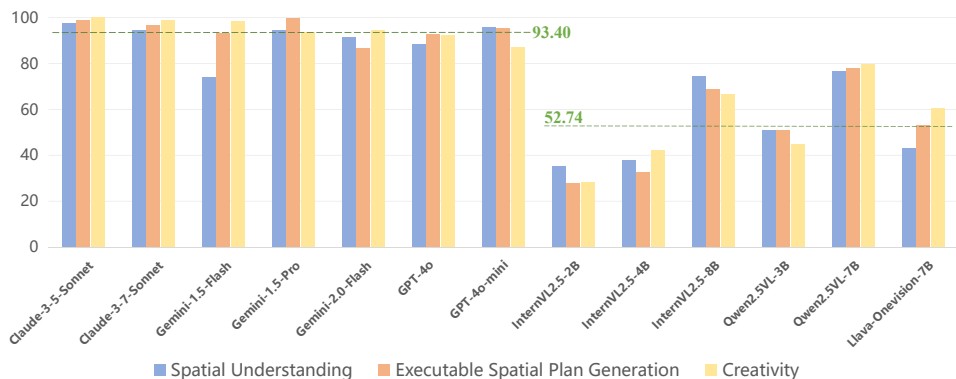

Figure 4: Bar chart of Output Success Rate (OSR) for MLLMs. Two green dotted lines indicate the average OSR of proprietary models and open-source models, respectively.

GPT-4o-mini, obtains merely 26.7% accuracy. Some models that are more capable in general AI tasks, e.g. GPT-4o, perform worse than GPT-4o-mini, which indicates that our spatial reasoning task still remains challenging for most MLLM-based agents.

(5) **Spatial Commonsense**: We evaluated these agents on spatial commonsense tasks relevant to humans' daily life, including the rationality of location, *etc*. The results in Table 1 reveal that most proprietary models have abundant spatial commonsense, achieving comparable responses to human-annotated answers. Open-source MLLM-based agents also show competent performances though with marginally lower consistency scores.

**Output Success Rate.** We statistically calculate the proportion of these MLLM-based agents that successfully respond with executable plans. In the Figure 4, we can observe that for the majority of proprietary models, their instruction-following capability and comprehension of 3D data are relatively strong, thus they can generate the corresponding executable blueprints according to their spatial planning. Gemini-1.5-Flash scores 73.81 on OSR which is below the mean line of 93.40 due to its incorrect understanding of the output dimension of the executable plan. The average line of open-source MLLM-based agents is quite lower than that of proprietary models, revealing that these agents are only effective for basic visual or textual understanding, while further training is still required for these 3D spatial tasks, such as spatial planning. Full metrics of Figure 4 are provided in the Supplementary Material.

**Planning Output Visualization.** We visualize some planning results in Figure 5. We can find that for some relatively easier tasks, most agents with strong capabilities can achieve great performance similar to the structure in the reference image. For example, as shown in Figure 5, agents are required to *build a potted tree with azalea flowers*, and Claude-3.7-Sonnet and Gemini-1.5-Pro show effective results under the tasks of spatial understanding and executable spatial plan generation, respectively. For the creativity task, the agent accessing GPT-4o can analyze and plan what blocks should be utilized and in what form to combine sub-structures into an integral whole. Moreover, the agent tend to consider how to increase the diversity and creativity of the overall structure and appearance, revealing its capabilities of spatial intelligence. More visualization results of all tasks are provided in the Supplementary Material.

**Failure Cases Analysis.** We provide some failure cases in Figure 5. We can observe that there are several causes of failure, which leads to agents being unable to generate the executable results based on their planning. For some low-parameter open-source MLLM-based agents, they have difficulty handling the 3D executable structures well, generating repetitive or confusing blueprints, leading to compilation failure. Some powerful proprietary models can understand some requirements and build the substructures, but there is still a spatial misunderstanding in their planning of combining them. For example, Claude-3.5-Sonnet wrongly overlaps the five rings of the Olympics Rings instead of laying them flat on the same plane, which is not in line with commonsense. For most MLLM-based agents, there is usually a severe implementation gap, i.e., they can not convert their textural planning into a spatial structure, which is precisely their huge defect in spatial planning. More visualization results of failure cases are provided in the Supplementary Material. Moreover, we provide deep analyses of the failure reasons for the cases with a summarization as follows:

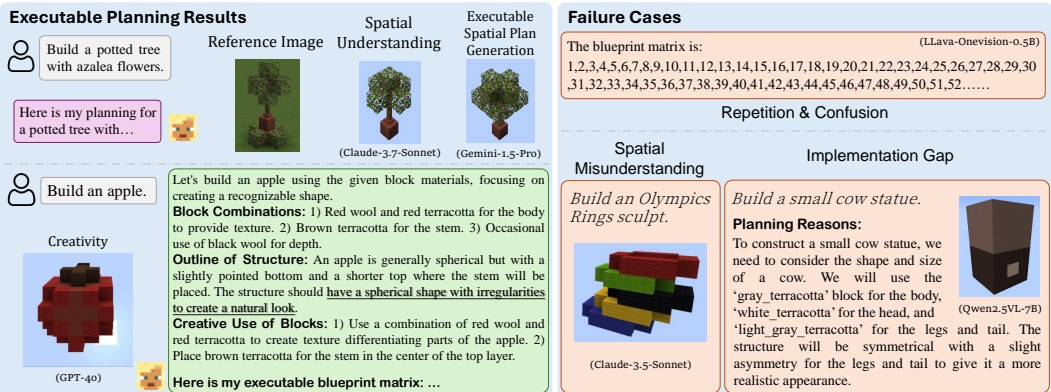

Figure 5: Visualization of executable planning results (left) and failure cases (right).

**(1) Spatial Misunderstanding:** Agents frequently misinterpret 3D positional relationships or fail to maintain the correct spatial arrangements, which highlights a persistent weakness in spatial grounding and planning.

**(2) Implementation Gap:** The agents have a central issue that they can not transform high-level textual plans into precise and executable blueprint matrices. The integration of substructures often fails due to incorrect block indexing, orientation errors or inconsistent spatial logic, leading to blueprint parsing or execution failures. This is essentially because the model's understanding of data structures such as codes or DSL and 3D matrices from a numerical perspective is still limited. If these models strengthen the training of spatial data, it may enhance the capabilities of these agents.

**(3) Structural Degeneration under Complexity:** When the tasks demand non-cubic, asymmetric or creative designs, the agents tend to collapse into simple and box-like outputs or disorganized results. This indicates that their limited ability to scale from basic patterns to more abstract and complex architectural concepts.

These failure modes reflect deeper limitations in MLLM's capabilities to perform hierarchical spatial planning, maintain geometric consistency and ground language into manipulable 3D structures. They also provide more research directions for MLLMs, e.g., to improve multi-modal spatial understanding, align linguistic abstraction with executable plans or enhance agent's ability for structural composition in open-ended 3D environments.

## 4 Related Works

**Spatial Intelligence.** Spatial intelligence involves thinking about the shapes and arrangements of objects in space and about spatial processes, such as the deformation of objects, and the movement of objects and other entities through space. Current works mainly focus on spatial understanding and spatial reasoning [8, 9, 10, 11, 40, 41, 42, 43, 44, 45]. VSI-Bench [8] first introduces the definition of visual-spatial intelligence and proposes a benchmark for it. SpatialVLM [9] presents an automatic framework generating millions of VQA samples of spatial reasoning for VLMs' evaluation. Lego-Puzzles [11] introduces a scalable benchmark with several VQA samples including tasks in multi-step spatial reasoning. However, these benchmarks suffer from the gap between abstract spatial understanding and concrete task execution. In this paper, we introduce an innovative benchmark concentrating on spatial planning, where the open-world AI agents need to generate executable spatial plans based on its spatial perception and cognition for architecture and indoor decorations. We also introduce diverse evaluation dimensions such as creativity and spatial commonsense to realize a comprehensive assessment for spatial planning capabilities.

**Minecraft for AI Research.** Minecraft is a 3D world sandbox video game with diverse game mechanics supporting various tasks and activities. Benefiting from its open-ended property, the training and evaluation of autonomous agents built on Minecraft are quite inspiring for the research in the field of artificial intelligence and embodied AI. There are several related works [12, 46, 47, 13, 48, 49, 50, 51, 52] contributing to the development in recent years. VPT [47] utilizes Youtube videos for agents' large-scale pretraining. MineDojo [12] features a massive database collected

automatically from the Internet and learns a MineCLIP model by watching thousands of Youtube videos. Voyager [13] imitate behavior by pseudo-labeling actions by plugging GPT-4 while Optimus-2 [49] learned a VLA-based model with MLLMs for high-level planning. These works are merely confined to traditional embodied planning tasks like skill learning or tech-tree goals. Compared to these works, we propose a new benchmark MineAnyBuild to evaluate AI agents in spatial intelligence, which is an emerging research field regarding the ability of AI agents to reason about 3D space.

## 5  Conclusion

We introduce MineAnyBuild, an innovative benchmark designed to evaluate spatial planning for open-world AI agents. Our MineAnyBuild consists of 4,000 curated tasks with 500+ buildings and decoration assets for evaluating spatial planning, and approximately 2,000 VQA pairs for spatial reasoning and commonsense evaluation. Extensive experiments on 13 advanced MLLM-based agents reflects that there is still a great growth space for spatial intelligence of agents. We believe that our MineAnyBuild will pioneer a novel paradigm to evaluate spatial intelligence, while advancing the development of open-world AI agents with spatial planning capabilities.

## 6  Acknowledgments

This work is supported by National Key Research and Development Program of China (2024YFE0203100), National Natural Science Foundation of China (NSFC) under Grants No.62476293, National Postdoctoral Program for Innovative Talents under Grant Number BX20250379, China Postdoctoral Science Foundation under Grant Number 2025M771521, and General Embodied AI Center of Sun Yat-sen University.

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
