# OpenReview forum: "MineAnyBuild: Benchmarking Spatial Planning for Open-world AI Agents"
_NeurIPS.cc/2025/Datasets_and_Benchmarks_Track — NeurIPS 2025 Datasets and Benchmarks Track poster_

### Official Review · Reviewer_moVA · 2025-06-01

**Rating:** 4
**Confidence:** 4

**Summary:**

The paper introduces MineAnyBuild, a large-scale benchmark designed to evaluate the spatial-planning abilities of open-world AI agents within the game Minecraft. The benchmark includes 4,000 tasks that assess five core skills: Executable Spatial Plan Generation, Spatial Understanding, Spatial Reasoning, Creativity, and Spatial Commonsense. In addition to these tasks, the benchmark features around 2,000 visual question answering (VQA) pairs modeled on mental-rotation challenges and commonsense scenes to further complement plan-generation tasks. To construct the dataset, the authors implement a three-stage, “infinitely expandable” data-curation pipeline that mines player-generated content, filters it, and applies annotations. Several state-of-the-art multimodal large language model (MLLM) agents are benchmarked.

**Dataset Code Accessibility:**

Partly

**Dataset Code Comments:**

The submission provides the dataset and code in a readily accessible and well-documented format, although the code is not complete by now.

**Ethical Comments:**

No significant ethical concerns remain. The benchmark does not involve human subjects, sensitive personal data, or real-world deployment risks.

**Ethical Considerations:**

No, there are no or only very minor ethics concerns

**Final Justification:**

Overall, I believe the benchmark makes a meaningful and timely contribution by shifting focus from passive VQA-style evaluation to execution-based spatial reasoning in a complex 3D world. This work would benefit from stronger baselines and deeper analysis. Given the benchmark's novelty, scale, and solid foundations, I maintain my borderline accept recommendation. I believe it will be of interest to the embodied AI and multimodal communities and can serve as a valuable resource with further refinement.

**Limitations Weaknesses:**

- There is no justification for why different skills (e.g., creativity, commonsense) receive their assigned importance (e.g. 0.8, 0.05).
- Although each task includes a *difficulty_factor*, the paper does not analyze how difficulty affects performance or break down baseline results across difficulty tiers, limiting insights into model failure patterns.
- Around half of the task instructions are machine-generated, yet the paper offers minimal information about quality control processes, lacking metrics like rejection rates, inter-annotator agreement, or edit statistics.
- The baseline evaluation excludes human experts and domain-specific and reinforcement learning-based Minecraft agents, missing an opportunity to demonstrate how well the benchmark captures planning abilities beyond language modeling.

**Strengths Contributions:**

- Addresses an under-explored capability for embodied agents by moving beyond VQA-style benchmarks to execution tasks.
- Substantially larger and broader than prior Minecraft datasets, with 4,000 building tasks and 2,000 VQA pairs covering perceptual, reasoning, and creative dimensions.
- Introduces an expandable, automated data-curation pipeline (scraping → filtering → annotation) that enables continuous dataset growth without manual crowdsourcing.
- Provides a thorough baseline study using 13 proprietary and open-source models in zero-shot settings, with both quantitative metrics and qualitative analyses.
- Presents results clearly through informative figures and a compact summary table, making the paper well-structured and easy to follow.

---

> ### Author Rebuttal · Authors · 2025-07-31
>
> Thank you for your acknowledgement that: 1) our benchmark addresses an under-explored capability for embodied agents, 2) our datasets are substantially larger and broader than prior ones, 3) our data curation pipeline is expandable, 4) our experiments are clearly presented with informative figures and tables, 5) our paper is well-structured and easy to follow. We respond to all the issues you pointed out in detail below. We hope that our response will match your expectations.
>
> > **Q1.** There is no justification for why different skills (e.g., creativity, commonsense) receive their assigned importance (e.g. 0.8, 0.05).
> >
>
> We assign different importances to each dimension of a specific task based on the following principle: we first select a main scoring dimension based on **human preference priors** and assign it a relatively high importance. For the remaining dimensions, we evenly distribute the remaining weight proportion to **ensure the output buildings have basic structures without obvious errors**.
>
> Taking Creativity task as an example, we set the weight of Creativity dimension to 0.8, while assigning lower weights 0.05 each to the remaining dimensions, including: 1) Completeness, 2) Complexity, 3) Architecture Structure, 4) Overall Aesthetic, Atmosphere and Fidelity.
>
> The assignment of importance can achieve a relatively good balance between the main evaluation dimension of creativity task and other basic scoring dimensions. We will add these justifications into our revised manuscript.
>
> > **Q2**. Although each task includes a *difficulty_factor*, the paper does not analyze how difficulty affects performance or break down baseline results across difficulty tiers, limiting insights into model failure patterns.
> >
>
> The original intention of setting the *difficulty_factor* was to classify the difficulty tiers and further evaluate MLLM-based agent's performance on tasks of different difficulty levels. However, considering the limited time, we do not test the full evaluation results under the corresponding difficulty tiers when writing the paper.
>
> As shown in Figure 18 of Supplementary Material, our difficulty factor distribution is an approximately normal (Gaussian) but right-skewed distribution. Based on the distribution, we define three difficulty tiers using the mean $\mu=2.8$ and standard deviation $\sigma=1.0$. Tasks with difficulty_factor $\leq1.8$ are considered **Easy**, those between 1.8 and 3.8 are **Medium**, and those $\geq3.8$ are **Hard**, following the rule of $\mu\pm\sigma$.
>
> Here is an simple experiment that we sample 50 data and evaluate a baseline model (GPT-4o) in Creativity task with these difficulty tiers. The model achieves a score of 4.35 on Easy tasks, 2.95 on Medium tasks and 0.10 on Hard tasks. The result indicates that for easy tasks, agent can achieve a good score, while for hard tasks (e.g., to build an extremely complex house), it cannot complete correct execution and obtain a very low score.
>
> We will incorporate more details into of our revised Supplementary Material.
>
> > **Q3.** Around half of the task instructions are machine-generated, yet the paper offers minimal information about quality control processes, lacking metrics like rejection rates, inter-annotator agreement, or edit statistics.
> >
>
> The task instructions generated by MLLMs are ~20%, which are mainly the instructions in Executable Spatial Plan Generation task. For Creativity, Spatial Understanding and Spatial Reasoning tasks, we provide templates to generate instructions by rules.
>
> You can find the example of templates for Creativity task in Figure 10 of Supplementary Material. The instructions for Spatial Understanding task are automatically generated by the coordinates of each block in the architecture data, and those for Spatial Reasoning task are selected from 3 predefined questions. We manually annotated the instructions and corresponding reference answers for Spatial Commonsense task.
>
> We will provide more details on our data curation pipeline in our revised Supplementary Material. We reanalyze our data generated before and provide the lacking information and metrics including rejection rates and inter-annotator agreement about quality control processes for Executable Spatial Plan Generation task as follows.
>
> **1) For rejection rates:** We randomly sample 100 instructions from this task and observe **a rejection rate of** **11%**, suggesting that most machine-generated instructions are broadly acceptable with minimal invalid cases. In most of the rejected cases, information of the instructions is either too redundant or contains some extraneous information. Thanks to the powerful instruction generation ability of GPT-4.1, the quality of most generated instructions is already quite high. Our manual processing is just to ensure that they meet our expectations.
>
> **2) For inter-annotator agreement (IAA):** We randomly select 50 instructions from this task and ask two annotators to judge whether each instruction is acceptable/rejected. We calculate Cohen’s Kappa to evaluate IAA and obtain a score of $\kappa=0.63$, indicating a strong level of agreement and the reasonably high quality of machine-generated task instructions.
>
> We will incorporate the information into our revised Supplementary Material.
>
> > **Q4.** The baseline evaluation excludes human experts and domain-specific and reinforcement learning-based Minecraft agents, missing an opportunity to demonstrate how well the benchmark captures planning abilities beyond language modeling.
> >
>
> Our manuscript mainly focuses on evaluating MLLM-based agents, which is highly representative for the evaluation of AI agents. The baselines including human experts, domain-specific and reinforcement learning-based agents are well worthy of being discussed and researched in the future. Due to several engineering challenges of code implementation and tight time constraints, we provide some transferring insights below for future research:
>
> **1) For human experts:** We plan to recruit ~10 human participants with experience in Minecraft building by referencing current work [1], to perform the same planning tasks to build the architectures. We will utilize the same evaluator and evaluation criteria to assess human experts and agents, so as to provide more comprehensive evaluation results.
>
> **2) For domain-specific agents:** Domain-specific agents, such as Voyager [2], could potentially perform well by leveraging task-specific knowledge or predefined strategies on Minecraft tech-tree tasks. The input/output of these agents cannot be directly transferred to our benchmark, therefore we need to do some adaptation works for further evaluation. Constructing some in-domain agents for our building data leaves some future works to better improve the intelligence of AI agents.
>
> **3) For reinforcement learning-based agents:** In Section F.1 of Supplementary Material and the GitHub repository, we have stated that we plan to develop RL-based agents under popular RL frameworks (e.g., Mineflayer [3], MineDojo [4], and MineRL [5]) in the future. These agents will adopt an architecture similar to Vision-Language-Action (VLA) models. These VLA models will focus on how to perform action modeling on vision and language inputs and embeddings, which will bring certain help to the research of AI agents other than language modeling. We believe that this research will also be helpful for the research in Embodied AI in the future.
>
> [1] Jiang, Yunfan, et al. "Behavior robot suite: Streamlining real-world whole-body manipulation for everyday household activities." *arXiv preprint arXiv:2503.05652* (2025).
>
> [2] Wang, Guanzhi, et al. "Voyager: An open-ended embodied agent with large language models." *arXiv preprint arXiv:2305.16291* (2023).
>
> [3] PrismarineJS. "Mineflayer." https://github.com/PrismarineJS/mineflayer.
>
> [4] Fan, Linxi, et al. "Minedojo: Building open-ended embodied agents with internet-scale knowledge." *Advances in Neural Information Processing Systems* 35 (2022): 18343-18362.
>
> [5] Guss, William H., et al. "Minerl: A large-scale dataset of minecraft demonstrations." *arXiv preprint arXiv:1907.13440* (2019).
>
> > **Dataset Code Comment:** The submission provides the dataset and code in a readily accessible and well-documented format, although the code is not complete by now.
> >
>
> In our GitHub repository, only the code for Data Curation Pipeline hasn't been released due to some temporary additional work at that time. The main code for evaluation has been released with documentation, which is basically sufficient for evaluating our benchmark.
>
> As mentioned in this year's policies for the rebuttal submission, we will not update our code during the rebuttal period. Therefore, we will update our incomplete code and follow the plan we listed in our GitHub repository for future updates.

---

### Official Review · Reviewer_wiuT · 2025-06-30

**Rating:** 5
**Confidence:** 4

**Summary:**

This paper proposes MineAnyBuild benchmark, designed to evaluate spatial planning for open-world AI agents. It consists of 4,000 curated tasks with 500+ buildings and decoration assets for evaluating spatial planning, and approximately 2,000 VQA pairs for spatial reasoning and commonsense evaluation.

**Dataset Code Accessibility:**

Yes

**Dataset Code Comments:**

The github seems to work well with most information available.
https://github.com/MineAnyBuild/MineAnyBuild

**Ethical Considerations:**

No, there are no or only very minor ethics concerns

**Final Justification:**

This is a timely work for evaluating the spatial intelligence. My concerns have clearly addressed through the rebuttal. Thus, I vote for acceptance for this paper.

**Limitations Weaknesses:**

W1. Are the results in Table1 derived from mutiple times of experiments? such as average results based on 10 API calls?

MLLM can generate different outputs even with the same prompt, the authors should run multiple times of the experiment. Now I can only see GPT-4.1 on Creativity task was run 10 times as shown in F.3.3 Error Bar Chart of Scoring by Critic Models in Supplementary file.

W2. Some larger open-source models should be definitely included in the evaluation, then there will be some different conclusions from the present. Stronger GPU should be used other than 3090 or 4090.

W3. More open-source MLLM models should  be included [1].
[1] A Survey on Multimodal Large Language Models. https://arxiv.org/abs/2306.13549

W4. Deeper insights and analyses on the failure cases would be very very meaningful and valuable for the community. Basically the authors are just describing what the MLLM did poorly, rather than the deeper reasons/thoughts.

W5. In Table 1, the Spatial Reasoning score of GPT-4o (24.4) and GPT-4o-mini (26.7), whcih is not so reasonable to me. Please kindly explain on this.

**Strengths Contributions:**

S1. Spatial planning is a crucial and emerging topic in the field of AI, thus this is a very timely benchmark for this task.

S2. This paper is well written and easy-to-follow, having high potential to draw attention from broad AI researchers.

S3. The evaluation task based on Minecraft is properly defined and designed.

---

> ### Author Rebuttal · Authors · 2025-07-31
>
> We are grateful for your recognition that: 1) our benchmark is very timely for the emerging topic in AI fields, i.e., spatial planning, 2) our paper is well-written and easy to follow, 3) our evaluation tasks are properly defined and designed. We respond to all the questions you raised in detail below. We hope our response will address your concerns.
>
> > **Q1.** Are the results in Table 1 derived from mutiple times of experiments? such as average results based on 10 API calls? MLLM can generate different outputs even with the same prompt, the authors should run multiple times of the experiment.
> >
>
> During the experimental process, we find that the average results obtained from multiple runs are **actually quite similar** to those from a single call, and the improvement in accuracy brought by multiple calls is very marginal. Therefore, **considering the time consumption and economic cost** brought by multi-run evaluation, we conduct a single run as the final test result.
>
> > **Q2.** Some larger open-source models should be definitely included in the evaluation, then there will be some different conclusions from the present. Stronger GPU should be used other than 3090 or 4090.
> >
>
> Thanks for your comments. The evaluation of open-source models with larger parameters is very important for our benchmark. We have stated that we would supplement it in the future in Section F.2 of Supplementary Material.
>
> We are sorry that it is difficult for us to urgently collect more GPU resources to test these larger open-source models during the rebuttal period. Considering the page limit of our paper, we will update the evaluation results of some larger open-source models in our GitHub repository as far as possible.
>
> > **Q3.** More open-source MLLM models should be included [1]. [1] A Survey on Multimodal Large Language Models. https://arxiv.org/abs/2306.13549
> >
>
> Thanks for your suggestions. As we answered in Q2, we will update more open-source MLLMs in the future to improve our evaluation and benchmark in our GitHub repository. We plan to evaluate these following open-source Multi-modal Large Language Models in the future: Llama-3.2-Vision, InternVL3, LLaVA-1.6, Qwen-VL-Max, Molmo, etc.
>
> > **Q4.** Deeper insights and analyses on the failure cases would be very very meaningful and valuable for the community. Basically the authors are just describing what the MLLM did poorly, rather than the deeper reasons/thoughts.
> >
>
> Thanks for your valuable suggestions for improving our paper. For failure cases in Section 3.3 (Figure 5) of the main text and Section G.2 (Figure 21) of Supplementary Material, we add the deep analyses for the failure reasons with a summarization as follows:
>
> **1) Spatial Misunderstanding:** Agents frequently misinterpret 3D positional relationships or fail to maintain the correct spatial arrangements, which highlights a persistent weakness in spatial grounding and planning.
>
> **2) Implementation Gap:** The agents have a central issue that they can not transform high-level textual plans into precise and executable blueprint matrices. The integration of substructures often fails due to incorrect block indexing, orientation errors or inconsistent spatial logic, leading to blueprint parsing or execution failures. This is essentially because the model's understanding of data structures such as codes or DSL and 3D matrices from a numerical perspective is still limited. If these models strengthen the training of spatial data, it may enhance the capabilities of these agents.
>
> **3) Structural Degeneration under Complexity:** When the tasks demand non-cubic, asymmetric or creative designs, the agents tend to collapse into simple and box-like outputs or disorganized results. This indicates that their limited ability to scale from basic patterns to more abstract and complex architectural concepts.
>
> These failure modes reflect deeper limitations in MLLM's capabilities to perform hierarchical spatial planning, maintain geometric consistency and ground language into manipulable 3D structures. They also provide more research directions for MLLMs, e.g., to improve multi-modal spatial understanding, align linguistic abstraction with executable plans or enhance agent’s ability for structural composition in open-ended 3D environments.
>
> We will incorporate these analyses into our revised manuscript and Supplementary Material.
>
> > **Q5.** In Table 1, the Spatial Reasoning score of GPT-4o (24.4) and GPT-4o-mini (26.7), which is not so reasonable to me. Please kindly explain on this.
> >
>
> Thanks for your comment. We actually evaluated these results, but we missed explanations for the analysis of the ”abnormal“ results in the Supplementary Material. The evaluation results of other models (e.g., GPT-4o) are normal, while the output results of GPT-4o-mini are **abnormally uniform**. Due to the smaller parameters and weaker capabilities of GPT-4o-mini, most of the outputs are the frequent answers (e.g., B/C/False with 765/640/288 times), and A/D/True (with 10/24/0 times) are rarely seen. It shows that GPT-4o-mini is more likely to “guess” a relatively higher accuracy in this test but the absolute score is not high, though our data is evenly and randomly distributed across all options. In contrast, stronger models may attempt to reason more deeply, which may not always get higher scores on difficult questions.
>
> We will incorporate these explanations in our revised Supplementary Material.

---

> > ### Comment · Reviewer_wiuT · 2025-08-04
> > **Thanks for your kind rebuttal.**
> >
> > Thanks for your kind rebuttal. Most of my concerns have been addressed. Good and timely work!

---

> > > ### Author Response · Authors · 2025-08-04
> > > **Official Comment by Authors**
> > >
> > > Thank you very much for your positive feedback and for taking the time to review our responses. We are glad that our explanations addressed your concerns. Your constructive comments are greatly helpful in improving our work. We will incorporate these valuable suggestions into our modified version.

---

### Official Review · Reviewer_tETG · 2025-07-02

**Rating:** 5
**Confidence:** 5

**Summary:**

This paper proposes a novel benchmark, called MineAnyBuild, to evaluate the spatial planning ability of open-world AI agents. Rather than previous benchmarks constructed for the spatial intelligence evaluation which primarily adopt the VQA forms, the proposed MineAnyBuild requires the agent to generate executable architecture building plans, leading to a mitigation between abstract spatial understanding and concrete task execution. Diverse core evaluation dimensions are introduced in MineAnyBuild, e.g., spatial understanding, spatial reasoning, creativity, and spatial commonsense. Multiple existing MLLM-based AI agents are tested on the proposed benchmark, and the results reveal severe limitations of current AI agents in the proposed spatial planning tasks.
Contributions:
- A new benchmark for spatial planning evaluation of open-world AI agents based on the Minecraft game.
- Extensive verification experiments for existing MLLM-based AI agents on the spatial planning ability.
- An infinitely expandable data curation pipeline to benefit the spatial planning training and evaluation for open-world AI agents.

**Dataset Code Accessibility:**

Yes

**Ethical Considerations:**

No, there are no or only very minor ethics concerns

**Final Justification:**

I am very satisfied with the reply and recommend accepting it.

**Limitations Weaknesses:**

1. The manuscript proposes an interesting benchmark for a comprehensive evaluation for the spatial planning ability of open-world AI agents. However, there are some clarification issues that impact the understanding of the implementation and design principles. Firstly, in Section 2, there lacks a concrete description about the model’s output form, i.e., the blueprint matrix. Moreover, the authors claim that it is hard for the spatial reasoning and spatial commonsense tasks to present in spatial planning forms while not analyzing the reasons.
2. Section 3.2 Evaluation Metrics also causes some confusion. In particular, it seems that the evaluation for each task is separate, why do the authors calculate a weighted “Evaluation score” for different tasks? What is the relationship between “Evaluation score” and “Score (out of 10)”? What does the “overall” in Table 1 mean and what is the maximum value for “overall”? I suggest the authors refine the definitions carefully to facilitate understanding for the proposed evaluation metrics.
3. There lacks the statistics of the building types in the proposed benchmark. Moreover, the numbers of different task data do not seem to be provided as well.
4. The authors utilize the critic models to evaluate the proposed spatial planning task for AI agents. How to ensure the reliability and reasonability of the critic model, especially for the Creativity task? It would be better to provide an assessment for the reliability evaluation for the critic model. Since the MLLM model serves as the critic model, it is also important to analyze the possible noise brought by it to impact the performance evaluation.

As mentioned in the Weaknesses, I would like to see:
1. Clarifications for the model’s output forms, reason of designing discriminative forms for different task, and the evaluation strategy.
2. Missing details about the dataset.
3. The reliability assessment and explanation for the critic model.

**Strengths Contributions:**

1. The proposed MineAnyBuild focuses on an important but unexplored point in spatial intelligence, i.e., spatial planning, which can mitigate the gap between abstract spatial understanding and concrete task execution. Multiple supporting dimensions are integrated into the proposed MineAnyBuild to enable a comprehensive evaluation for the spatial planning ability of AI agents.
2. The paper is well-written and easy to follow. Figures and tables are nicely presented.
3. Supplementary materials are provided with plenty of details about dataset construction, task descriptions, evaluation strategies, and additional experimental results.
4. Visualization of failure cases are provided to intuitively analyze the problems of existing MLLM-based agents in spatial planning.

---

> ### Author Rebuttal · Authors · 2025-07-31
>
> We sincerely appreciate your acknowledgement that: 1) our MineAnyBuild focuses on important research fields and has a comprehensive evaluation for AI agents, 2) our paper is well-written and figures/tables are nicely presented, 3) Supplementary Material is provided with details and the visualization well analyze the problems of agents in spatial planning. We respond to all the comments you suggested in detail below. We hope our response will meet your expectations.
>
> > **Q1.** There are some clarification issues that impact the understanding of the implementation and design principles.
> >
>
> Thanks for your valuable suggestions for helping us improve our manuscript. We provide some explanations for the proposed clarification issues below.
>
> 1. **concrete description about the model’s output form.**
>
>     We have provided some examples in the task prompts (in Section E.1 of Supplementary Material), where we present some model’s output forms, e.g., blueprint matrix and planning. Due to our negligence, we actually omitted the descriptions about the output form in our paper. The following are the main details on the blueprint matrix and its post-processing:
>
>     ***a) output format of blueprint matrix:***
>
>     *The blueprint matrix has three dimensions. The three dimensions of this matrix are height (y-axis), depth (z-axis) and width (x-axis), respectively. The size of this matrix is determined by the sizes of the three dimensions.*
>
>     *The elements of the blueprint matrix are integers, each of which corresponds to a block. We use integers instead of strings as a representation of a block, to avoid an explosion in the numbers of tokens caused by a large number of elements. Therefore, we adopt a format similar to sparse matrices and use integers to represent elements of the matrix.*
>
>     ***b) output post-processing for architecture generation:***
>
>     *We use the list "block_materials" to represent the types of blocks that will be used in the building task. When querying the MLLM-based agents, we convert this list into a hash (a dictionary in Python) to indicate the mapping relationship between block types and integers. The “air” block is set to -1 by default. The integers corresponding to the block types are calculated by adding 1 to the index of this block type in the list. e.g., for the list [”grass”, “oak_wood”], the hash should be {”air”: -1, “grass”: 1, “oak_wood”: 2}. We also use the reversed hash during the execution process to establish a reverse mapping for further evaluation and visualization.*
>
>     These details are also reflected in the released codes and corresponding annotations. We will provide more details on the model’s output form in our revised Supplementary Material.
>
> 2. **reason why not present spatial reasoning and spatial commonsense tasks in spatial planning forms.**
>
>     Our MineAnyBuild datasets mainly include three tasks, i.e., Executable Spatial Plan Generation, Spatial Understanding, and Creativity to **comprehensively evaluate the agents'  spatial planning ability in the aspects like instruction following or abstract architecture  understanding**. Beyond these tasks, we additionally supplement the VQA tasks of Spatial Reasoning and Spatial Commonsense for the intention of assisting the evaluation for the agents' capabilities in **commonsense-based spatial reasoning**, which **significantly impacts its spatial planning accuracy**. Defining these two tasks with **the VQA form is convenient for capability evaluation, which is also demonstrated in other previous benchmarks**.
>
>     We will incorporate the explanation into our revised Supplementary Material for improving the clarity.
>
>
> > **Q2.** Section 3.2 Evaluation Metrics also causes some confusion. I suggest the authors refine the definitions carefully to facilitate understanding for the proposed evaluation metrics.
> >
>
> Thanks for your suggestions for improving our paper. Here are the missing details.
>
> 1. **the weighted “Evaluation score” for different tasks.**
>
>     The “Evaluation Scores” are the scores solely evaluated by critic model. **The “Evaluation Scores” are separated and designed to evaluate for each task.** During our experiments, we find that if we grade a single task from only one dimension, the scores of the critic model are not entirely reliable, making it hard to distinguish the fundamental differences among some outputs. Therefore, we consider introducing multiple scoring dimensions for evaluation. Inspired by Chain-of Thought, we also hope that multi-dimensional scoring can prompt the critic model to provide more accurate scores.
>
>     For example, in Creativity task, the weighted evaluation score includes five scoring dimensions for evaluation: 1) Creativity, 2) Completeness, 3) Complexity, 4) Architecture Structure, 5) Overall Aesthetic, Atmosphere and Fidelity. Taking "Completeness" as an example. If a building does not have a complete structure (e.g., incompleted or broken), it is meaningless to discuss its creativity. Therefore, we considered multiple dimensions to ensure the reliability of this scoring.
>
> 2. **relationship between “Evaluation score” and “Score (out of 10)”.**
>
>     In the experiments of our manuscript, we mainly focus on the “Evaluation Score” which is given by the critic model. We missed the clear explanations on these two scores. Taking Score for Spatial Understanding task as an example, the formula is:
>
>     $$Score=k_{1}*S_{Evaluation}+k_{2}*S_{Matching}$$
>
>     where $Score$ is the “Score (out of 10)” and the $S_{Evaluation}$ is the “Evaluation score”. $S_{Matching}$ is the matching score with a relatively low weight $k_2$, thus “Evaluation score” contributes more to “Score (out of 10)”. We will add more explanations about the notation and the score calculation in Section F.3.2 in our revised Supplementary Material.
>
> 3. **meaning of the “overall” and maximum value of it.**
>
>     The “Overall” Score is the average score of these five tasks, with a maximum value of 100. The calculation formula is:
>
>     $$S_{Overall}=\frac{(S_{ESPG}+S_{SU}+Acc_{SR}/10+S_{C}+S_{SC})}{5}*10$$
>     where $S_{ESPG}$, $S_{SU}$, $Acc_{SR}$, $S_{C}$ and $S_{SC}$ are the Score/Accuracy of Executable Spatial Plan Generation, Spatial Understanding, Spatial Reasoning, Creativity and Spatial Commonsense tasks.
>
>
> We will incorporate these details into our revised Supplementary Material. Thanks again for pointing out these missing details to help us improve our paper.
>
> > **Q3.** There lacks the statistics of the building types in the proposed benchmark. Moreover, the numbers of different task data do not seem to be provided as well.
> >
>
> Thank you for your suggestions and reminder. We will add the missing statistics with charts into our revised Supplementary Material. Briefly, there are the following points.
>
> - There are 24 building types in our proposed dataset, including house, fictional characters, items, etc.
> - The numbers of different tasks (including Executable Spatial Plan Generation, Spatial Understanding, Spatial Reasoning, Creativity and Spatial Commonsense) are 946, 473, 1728, 1419 and 50, respectively.
>
> > **Q4.** The authors utilize the critic models to evaluate the proposed spatial planning task for AI agents. How to ensure the reliability and reasonability of the critic model, especially for the Creativity task? It would be better to provide an assessment for the reliability evaluation for the critic model. Since the MLLM model serves as the critic model, it is also important to analyze the possible noise brought by it to impact the performance evaluation.
> >
>
> Thank you for your suggestions. Based on your suggestions, we provide some new statistics and analyses as follows.
>
> 1. **an assessment for the reliability evaluation.**
>
>     We provide some new experiments to present the reliability and reasonability of the critic model.
>
>     1. **Spearman correlation**
>
>         To evaluate the reliability of our MLLM-based critic model, we randomly sample 50 task outputs from our benchmark. We rate these outputs from 1 to 10 just like the critic model does. We compute the Spearman correlation between human and model scores, obtaining $\rho=0.76 (p<0.01)$. The value shows that the reliability and reasonability of the critic model.
>
>     2. **Intra-model Consistency**
>
>         We have provided an error bar chart of scoring by critic models in Section F.3.3 of Supplementary Material and its corresponding analysis. From this chart, we can observe that for most data points, the upper and lower error of the scores is concentrated within 1. The scores given by the critic model are relatively consistent, which also lays a foundation for the reliability of our benchmark.
>
> 2. **possible noise brought by critic model.**
>
>     We have simply discussed it by the error bar chart of scoring by critic models in Section F.3.3 of Supplementary Material. For analysis of the possible noise, we find that there are mainly two probable aspects.
>
>     **a) Different scoring comments:** We require the critic model to output the comments while giving the evaluation scores. As shown in the evaluation prompts in Section E.2 of Supplementary Material, we manually provide the output examples and the comments generated by critic model are possible sources of noise.
>
>     **b) Manually customized scoring tiers:** As shown in our evaluation prompts, we provided some words for ratings of 1, 5 and 10 respectively, as scoring references for the critic model. The manually-written prompts in this part may introduce some possible noise.
>
>     But overall, the impact of these two possible sources of noise on our evaluation is marginal, which can be supported by the provided error bar chart.
>
>
> We will incorporate these statistics and analyses into our revised Supplementary Material.

---

### Official Review · Reviewer_LTBC · 2025-07-03

**Rating:** 5
**Confidence:** 4

**Summary:**

The paper presents MineAnyBuild, a comprehensive benchmark designed to evaluate the spatial planning capabilities of open-world AI agents within the Minecraft environment. Unlike prior VQA-style benchmarks that mostly assess abstract spatial reasoning, MineAnyBuild focuses on executable architectural planning grounded in multi-modal human instructions. It covers four critical dimensions — spatial understanding, reasoning, creativity, and commonsense — with 4,000 curated tasks and a scalable data collection paradigm. Overall, this work highlights both the limitations and the promise of current MLLM-based agents in real-world spatial planning scenarios.

**Dataset Code Accessibility:**

Yes

**Dataset Code Comments:**

Dataset URL: https://huggingface.co/datasets/SaDil/MineAnyBuild
Code URL: https://github.com/MineAnyBuild/MineAnyBuild

**Ethical Considerations:**

No, there are no or only very minor ethics concerns

**Final Justification:**

Thanks for the authors' detailed response. After reading other reviewers' comments and authors' thorough response, I choose to raise my final rating.

**Limitations Weaknesses:**

1. The benchmark focuses solely on the Minecraft domain, which limits its ability to comprehensively evaluate spatial planning skills across diverse or real-world environments. Including data from non-game or real-world scenarios would improve its generalizability and practical impact.

2. The paper lacks sufficient statistical analyses of the built benchmark dataset, which are important for understanding the dataset’s diversity.

3. The experimental evaluation is somewhat limited, as it does not consider more diverse baseline models. Including chain-of-thought reasoning models or advanced world models trained on similar domains would better demonstrate the benchmark’s challenge and the capabilities of current approaches.

**Strengths Contributions:**

The paper is well-organized and clearly written, with detailed descriptions of the dataset construction process. The proposed MineAnyBuild benchmark covers diverse dimensions like spatial reasoning, creativity, and commonsense, and importantly narrows the gap between abstract reasoning and real executable planning. The scalable data pipeline leveraging player-generated content is also a strong point, and the thorough evaluation with visualized failure cases effectively highlights current limitations of existing MLLM-based agents.

---

> ### Author Rebuttal · Authors · 2025-07-31
>
> Thank you very much for your recognition that: 1) our paper is well-organized and clearly written, 2) our benchmark covers tasks with diverse dimensions and thorough evaluation, 3) our scalable data pipeline is a strong point.  We respond to all the weaknesses you pointed out in detail below. We hope our responses will address your concerns.
>
> > **Q1.** The benchmark focuses solely on the Minecraft domain, which limits its ability to comprehensively evaluate spatial planning skills across diverse or real-world environments. Including data from non-game or real-world scenarios would improve its generalizability and practical impact.
> >
>
> We select Minecraft as our domain because it provides a visually grounded 3D space that supports diverse and complex spatial planning tasks. While our MineAnyBuild does not cover all real-world scenarios, we aim to provide a scalable and flexible platform to benchmark emerging multi-modal agents in open-ended environments. The diversity in block types, the extensive freedom of interaction, and the standardized 3D coordinate system enable Minecraft to be an ideal environment for assessing the spatial reasoning and planning capabilities of large AI models.
>
> **1) For non-game scenarios:** We can migrate our benchmark to a similar non-game environment, such as Isaac Sim or AI2Thor simulators for 3D scene generation. This generation task requires a framework to retrieve assets from asset library (e.g., Objaverse[1]) and properly place them into an indoor scene. **How to ensure that the positions and rotations of all assets are correctly arranged is exactly a form of spatial planning**. This task is one of the cutting-edge research directions in recent years, and the motivation of this is similar to that of our benchmark.
>
> **2) For real-world scenarios:** How to perform sim2real transfer for our benchmark is also one of our interests. We actually watched a video on YouTube showing physical blocks with same appearance as those in the game (e.g., grass block), **which can be spliced together using the built-in magnets, just like toy blocks and LEGO bricks**. We plan to build a physical (real-world) version of our benchmark in the future, and study how agents trained in simulation can be deployed to manipulate these blocks in our real world to construct buildings.
>
> We recognize that expanding our benchmark to non-game or real-world scenarios is crucial for spatial planning evaluation. And we believe that our MineAnyBuild will provide meaningful references for benchmark design applicable to these scenarios.
>
> [1] Deitke, Matt, et al. "Objaverse: A universe of annotated 3d objects." *Proceedings of the IEEE/CVF conference on computer vision and pattern recognition*. 2023.
>
> > **Q2.** The paper lacks sufficient statistical analyses of the built benchmark dataset, which are important for understanding the dataset’s diversity.
> >
>
> We have already provided some details and analyses of our built benchmark and dataset in our Supplementary Material, including dataset construction, task descriptions, evaluation strategies, and additional experimental results (as Reviewer tETG mentioned in the Strengths Contributions). We present the analyses of our dataset’s diversity below:
>
> Our MineAnyBuild has 4000+ curated tasks covering several critical dimensions for evaluation of AI agents. We have curated **483 diverse architectures/assets** from thousands of data candidates for planning tasks. We also collected **10+ large scenes** for Spatial Commonsense task and generated **192 stimuli** from the original datasets. The quantity and diversity of our datasets can also be **infinitely expanded and scaled** through our data curation pipeline.
>
> We have provided visualization examples and details for our MineAnyBuild tasks in Section C.5.2 and Section D.1 of Supplementary Material. For example, in Figure 15 & 16, we presented the data details of Spatial Reasoning and Spatial Commonsense these two tasks, including expanding the data diversity by **performing mirror symmetry on the stimuli along the X/Y/Z axes to obtain reasoning data**, or establishing **multiple perspectives** for examining spatial commonsense.
>
> We will add these statistical analyses to our revised Supplementary Material, to better assist readers in understanding our benchmark.
>
> > **Q3.** The experimental evaluation is somewhat limited, as it does not consider more diverse baseline models. Including chain-of-thought reasoning models or advanced world models trained on similar domains would better demonstrate the benchmark’s challenge and the capabilities of current approaches.
> >
>
> Thanks for your suggestions and we fully recognize that evaluating more diverse baseline models could further strengthen our benchmark. In our evaluation prompts for MLLM-based agents, we **have already incorporated the Chain-of-Thought technique** to achieve better planning results. Moreover, we add the evaluation of some representative embodied models, **typically containing the design of CoT reasoning or world models**, in our Spatial Reasoning and Spatial Commonsense tasks as follows:
>
> | Embodied Models | Accuracy (Spatial Reasoning) | Score (Spatial Commonsense) |
> | --- | --- | --- |
> | Cosmos-Reason-1 [1] | 19.39 | 5.34 |
> | RoboBrain [2] | 21.64 | 5.14 |
> | RoboPoint [3] | 22.16 | 4.42 |
>
> By observing the results and comparing them with the existing results in our paper, these embodied models outperform most open-source MLLMs in Spatial Reasoning task, thanks to the high requirements for reasoning in the embodied tasks related to these models. However, these models perform poorly in Spatial Commonsense task, indicating that they still have deficiencies in understanding spatial relationships and architectural structures.
>
> [1] Azzolini, Alisson, et al. "Cosmos-reason1: From physical common sense to embodied reasoning." *arXiv preprint arXiv:2503.15558* (2025).
>
> [2] Ji, Yuheng, et al. "Robobrain: A unified brain model for robotic manipulation from abstract to concrete." *Proceedings of the Computer Vision and Pattern Recognition Conference*. 2025.
>
> [3] Yuan, Wentao, et al. "Robopoint: A vision-language model for spatial affordance prediction for robotics." *arXiv preprint arXiv:2406.10721* (2024).

---

> > ### Comment · Reviewer_LTBC · 2025-08-04
> >
> > Thanks for the authors' detailed response. After reading other reviewers' comments and authors' thorough response, I choose to raise my final rating to 5.

---

> > > ### Author Response · Authors · 2025-08-04
> > > **Official Comment by Authors**
> > >
> > > We are grateful for your positive feedback and for taking the time to review our responses. We are glad that our response meets your expectations and sincerely appreciate you raising the score. Your constructive comments are greatly helpful in improving our work. We will incorporate these valuable suggestions into our revised version.

---

### Note · Authors · 2025-08-12

We greatly appreciate the Area Chairs for organizing the excellent reviewing process. We would like to thank all reviewers for their precious time, insightful suggestions and valuable comments.

We are glad to see that our paper is appreciated for the following aspects:

- MineAnyBuild is a **timely** benchmark with **multiple evaluation tasks** for spatial intelligence, which is a **crucial and emerging** topic in AI research (Reviewers tETG and wiuT).
- Our benchmark constructs properly designed execution tasks, which importantly **narrows the gap between abstract reasoning and real executable planning** (All Reviewers LTBC, tETG, wiuT and moVA). Our paper also incorporates **comprehensive evaluation results** for various proprietary and open-source models (Reviewer moVA).
- MineAnyBuild introduces an **expandable and scalable** data curation pipeline (Reviewers LTBC and moVA).
- Our paper is **well-written, well-organized** and **easy to follow**, provided with **nicely presented figures/tables and visualizations** (All Reviewers LTBC, tETG, wiuT and moVA).

We have received constructive feedback in this rebuttal, and we summarize the main revisions as follows:

- **Missing details and clarifications**: We add the explanations for some missing details, including: a) concrete descriptions of the output form; b) the reason for two different forms of tasks; c) explanations for the proposed evaluation metrics; d) the abnormal results of GPT-4o-mini in Spatial Reasoning task; e) assigned importances for evaluation scores.
- **Deeper insights and analyses**: We deeply analyze the failure cases and possible noise brought by critic model. We also provide insights for transferring to non-game or real-world scenarios and adapting to domain-specific or reinforcement learning-based agents.
- **More statistical results and information**: We supplement statistics of our built benchmark. We also provide new statistics and analyses, including: a) assessment for the reliability evaluation; b) classifications of difficulty tiers based on *difficulty_factor*; c) metrics about quality control processes, e.g., rejection rate and inter-annotator agreement.

We will incorporate the above modifications into our revised paper and Supplementary Material for improving the quality and clarity. We hope the summary will help the Area Chairs and Reviewers recognize our contributions. We look forward to sharing our work with broad reasearch community of open-world AI agents.

---

### Decision · Program_Chairs · 2025-09-18

**Decision:**

Accept (poster)

**Comment:**

This work builds a benchmark for evaluating spatial planning abilities of open-world AI agents in the Minecraft game. Most reviewers considered this benchmark is valuable, as the spatial planning ability is important for agents, and it has not been well evaluated. There were some concerns, such as the limited environment in only Minecraft, limited evaluated models, lack of deep insights, etc. Although not all concerns are addressed, most reviewers still recognized this work. Overall, I think that although this work could be further improved, it could be accepted, mostly due to the importance of the studied task. The authors are encouraged to incorporate most important contents presented during the rebuttal into the final manuscript.